# Learning to Separate Domains in Generalized Zero-Shot and Open Set Learning: a Probabilistic Perspective

## Abstract

This paper studies the problem of domain division which aims to segment instances drawn from different probabilistic distributions. This problem exists in many previous recognition tasks, such as Open Set Learning (OSL) and Generalized Zero-Shot Learning (G-ZSL), where the testing instances come from either seen or unseen/novel classes with different probabilistic distributions. Previous works only calibrate the confident prediction of classifiers of seen classes (W-SVM Scheirer et al. (2014)) or taking unseen classes as outliers Socher et al. (2013). In contrast, this paper proposes a probabilistic way of directly estimating and fine-tuning the decision boundary between seen and unseen classes. In particular, we propose a domain division algorithm to split the testing instances into *known*, *unknown* and *uncertain* domains, and then conduct recognition tasks in each domain. Two statistical tools, *namely,* bootstrapping and Kolmogorov-Smirnov (K-S) Test, for the first time, are introduced to uncover and fine-tune the decision boundary of each domain. Critically, the uncertain domain is newly introduced in our framework to adopt those instances whose domain labels cannot be predicted confidently. Extensive experiments demonstrate that our approach achieved the state-of-the-art performance on OSL and G-ZSL benchmarks.

## 1 Introduction

This paper discusses the problem of learning to separate two domains which include the instances sampled from different distributions. This is a typical and general research topic that can be potentially used in various recognition tasks, such as Open Set Learning (OSL) and Generalized Zero-Shot Learning (G-ZSL). Particularly, OSL can break the constraints of the *closed set* in supervised learning, and aim at recognizing the testing instances from one of the seen classes (*i.e., known domain*), and the novel class (*i.e., unknown domain)*. The novel classes include the testing instances which have different distributions from that of the seen ones. In contrast, G-ZSL targets at distinguishing the labels of instances from the seen and unseen classes. Only the seen classes have the training instances, but unseen classes do not. Note that OSL does not explicitly give the class labels for those instances categorized as the novel class, but G-ZSL requires predicting the class labels of unseen classes. To address G-ZSL, semantic attributes or vectors are introduced as the intermediate representations; each (seen/unseen) class has one semantic prototype that contains class level information. Specifically, a reasonable solution of OSL and G-ZSL is via dividing the known and unknown domains. For training classes, the predictors are constructed to map visual features to the class label space (OSL), (or semantic space (G-ZSL)). Testing is performed on each separated domain to identify seen classes and the novel class (OSL), or both seen and unseen classes (G-ZSL).

The key question of OSL and ZSL is how to deal with the newly introduced novel class/unseen classes efficiently in the testing time. This is different from the conventional Zero-Shot Learning (ZSL) task which assumes that, in the testing stage, seen classes would not be misclassified as unseens, and vice versa; ZSL only uses the unseen classes for testing. Unfortunately, the predictors learned on training classes will inevitably make OSL or G-ZSL approaches tend to be biased towards the seen classes, and thus leading to very poor classification results for the novel class (OSL) or unseen classes (G-ZSL) Xian et al. (2017); Chao et al. (2016). We show an example in Fig. 1. On aPY dataset (described in Sec. 6.1) Farhadi et al. (2009), t-SNE van der Maaten & Hinton (2008) is

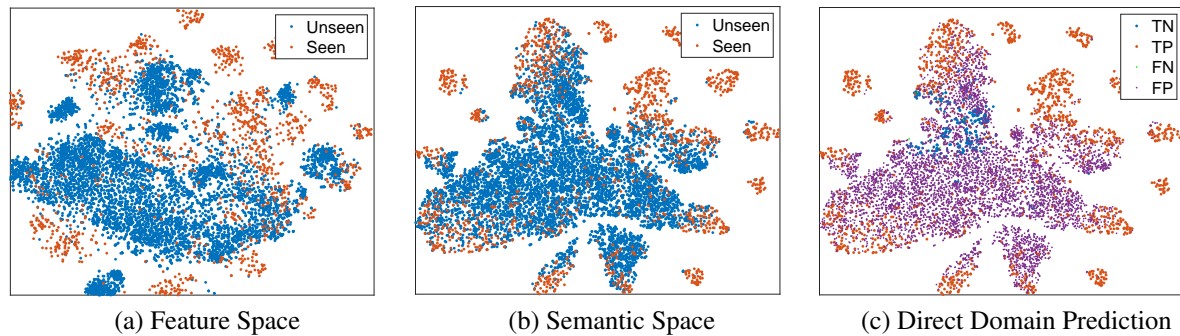

(a) Feature Space     (b) Semantic Space     (c) Direct Domain Prediction

Figure 1: **Feature and semantic spaces of seen and unseen classes.** We can see that a large of portion of instances from unseen classes is wrongly labeled as seen classes (FP). TN: True Negative; TP: True Positive; FN: False Negative; FP: False Positive.

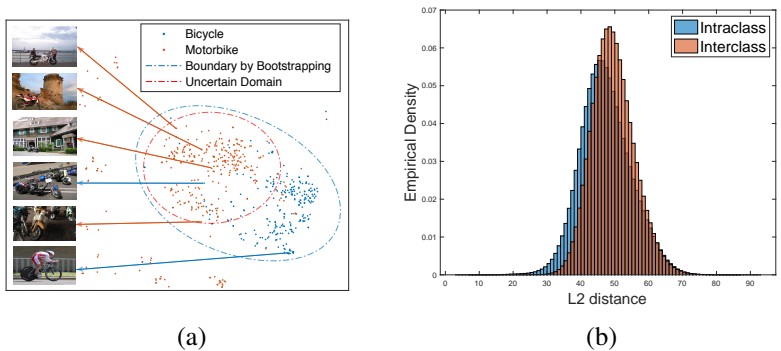

(a)             (b)

Figure 2: **(a) Illustration of Our Domain Division Algorithm:** The initial boundary of the known domain is estimated by bootstrapping. We can further divide an uncertain domain by K-S Test. Then we can recognize instances in each domain. **(b) The distribution of pairwise intraclass and interclass distances:** We compute the empirical density of the pairwise distance in aPY dataset (described in Sec. 6.1). There is a large overlapping of the distribution of the intraclass and interclass distances.

employed to visualize the distributions of the testing instances of the ResNet-101 features in Xian et al. (2017) (Fig. 1 (a)), and semantic features learned by SAE Kodirov et al. (2017) (Fig. 1 (b)). We categorize the SAE prediction as known or unknown domain labels and compare with the ground-truth in Fig. 1(c). We show that a large portion of unseen instances being predicted as one of the known classes.

A natural recipe for addressing this problem is to learn to separate domains by the distributions of instances; and different classifiers can be directly applied in each domain. However, there are still two key problems. First, visual features alone are not discriminative enough to help to distinguish the seen and unseen/novel classes. As Fig. 2 (a), *bicycle* and *motorbike,* respectively, are one of the seen and unseen classes[1] in aPY dataset (described in Sec. 6.1). We can observe that there is a large overlapping region between their t-SNE visualized feature distributions. That is, the visual features may not be representative enough to differentiate these two classes; the instances of *motorbike* (circled as the uncertain domain) may be taken as the *bicycle,* or vice versa; Second, the predictors trained on seen classes may be not trustworthy. A not well-trained predictor may negatively affect the recognition algorithms. Third and even worse, the performance of classifiers in each domain is still very sensitive to the results of domain separation: should the domain of one testing instance be wrongly divided, it would never be correctly categorized by the classifiers.

---

[1]Note that in OSL setting, we do not know the class label of *motorbike,* and its instances should be categorized as just the *novel* class.

To tackle the aforementioned issues, our key insight (see Fig. 2(a)) is to introduce a novel domain – *uncertain* domain that accounts for the overlapping regions of testing instances from seen or novel/unseen classes. Thus, the visual or semantic space can be learned to be divided into *known*, *unknown* and *uncertain* domains. The recognition algorithms will be directly employed in each domain. Nonetheless, how to divide the domains based on known information is also a non-trivial task. Though the supervised classifiers can learn the patterns of known classes, not all classes encountered during testing are known.

Formally, we propose exploiting the distribution information of seen and novel/unseen classes to efficiently learn to divide the domains from a probabilistic perspective. Our domain separation algorithm has two steps: the initial division of domains by bootstrapping, and fine-tuning by the Kolmogorov-Smirnov test. Specifically, according to extreme value theory Scheirer et al. (2014), the maximum/minimum confidence scores predicted by the classifier of each class can be taken as an extreme value distribution. Since we do not have the prior knowledge of the underlying data distributions of each class; bootstrapping is introduced here as an asymptotically consistent method in estimating an initial boundary of known classes. Nevertheless, the initial boundary estimated by bootstrapping is too relaxed to include novel testing instances as is illustrated in Fig. 2(b). To fine-tune the boundary, we exploit the K-S Test to validate whether the learned predictors are trustworthy in a specific region. The uncertain domain introduced thus accounts for those testing instances whose labels are hard to be judged. Recognition models can be conducted in each domain.

**Contributions:** The main contribution is to present a systematic framework of learning to separate domains by probabilistic distributions of instances, which is capable of addressing various recognition tasks, including OSL and G-ZSL. Towards this goal, two simple, most widely used, and very effective tools – bootstrapping and the Kolmogorov-Smirnov test, are employed to firstly initially estimate and then fine-tune the boundary. In particular, we introduce an uncertain domain, which encloses the instances which can hardly be classified into known or unknown with high confidence. We extensively evaluate the importance of domain division on several zero-shot learning benchmarks and achieved significant improvement over existing ZSL approaches.

## 2 RELATED WORKS

**One-Class Classification (OCC)**. It is also known as the unary classification or class-modeling. The OCC assumes that the training set contains only the positive samples of one specific class. By learning from such positive instances, OCC aims at identifying the instances belonging to that class. The common algorithms of OCC include One-class Support Vector Machine (OCSVM) SchÄlkopf et al. (2001), Local Outlier Factor (LOF) Breunig et al. (2000). OCSVM leverages Support Vector Data Description (SVDD) to get a spherical boundary in feature space. It regularizes the volume of hypersphere so that the effects of outliers can be minimized. The LOF measures the local deviation of the density of a given instance comparing to other instances, *namely* locality. The locality represents the density of the area. The instances in low-density parts can be taken as outliers. Note that all OCC algorithms just considered and build a boundary for the positive instances of one class.

**Open Set Learning** (OSL). It judges whether the instances belong to known/seen classes Sattar et al. (2015); Scheirer et al. (2014; 2013); Bendale & Boult (2015) or a novel unknown class. Both the OCC and OSL are able to divide the instances into known and unknown domains and recognize the known classes from the known domain. OSL aims at discriminating the instances into seen classes and instances beyond these classes are categorized into a single *novel* class. Critically OSL does not have the semantic prototypes of unseen classes to further give the class label of those instances in the novel class. Both the OCC and OSL are able to divide the instances into known and unknown domains and recognize the known classes in the known domain. Intrinsically, their key difference lies in whether leveraging the information of different seen classes in building the classifiers. Specifically, OCC only utilizes the instances of one class to learn its class boundary, whilst OSL can use the instances of different seen classes.

**Zero-Shot Learning (ZSL).** ZSL aims at recognizing the novel instances which have never been trained before. It transfers the knowledge learned from known source classes to recognize the testing instances from unknown target classes. The knowledge can be formulated as semantic attributes Farhadi et al. (2009); Lampert et al. (2009); Fu et al. (2015), semantic word vector Norouzi et al. (2013); Frome et al. (2013); Fu & Sigal (2016); Xu et al. (2017) or ontology Rohrbach et al. (2010).

However, ZSL usually assumes that the unseen classes cannot be mis-classified as seen classes and vice versa. This has greatly simplified the learning task.

**Generalized Zero-Shot Learning**. Chao *et al.* Changpinyo et al. (2016) realized that it is nontrivial to directly utilize the existing Zero-Shot Learning algorithms in a more general setting, *i.e.*, G-ZSL. In such a setting, the testing instances can come from either the seen or unseen classes. A thorough evaluation of G-ZSL is further conducted in Xian et al. (2017). Their results show that the existing ZSL algorithms do not perform well if directly applied to G-ZSL. The predicted results are inclined to be biased towards seen classes.

## 3 LEARNING TO CATEGORIZE INSTANCES INTO DIFFERENT DOMAINS

### 3.1 PROBLEM SETUP

In learning tasks, we are given the training dataset, *i.e.*, seen classes, of $n_s$ instances, $\mathcal{D}_s = \{\mathbf{x}_i, \mathbf{y}_i, l_i\}_{i=1}^{n_s}$: $\mathbf{x}_i \in \mathbb{R}^n$ is the feature of $i_{th}$ instance with the class label $l_i \in \mathcal{C}_s$, where $\mathcal{C}_s$ is the source class set; $n_s^c$ is the number of instances in seen class $c$. Analogous to standard ZSL setting, we introduce the target label classes $\mathcal{C}_t$ with $\mathcal{C}_s \bigcap \mathcal{C}_t = \emptyset$ and the total class label set $\mathcal{C} = \mathcal{C}_s \cup \mathcal{C}_t$. $\mathbf{y}_i$ is the semantic attribute vector of instance $\mathbf{x}_i$. In general, the $\mathbf{y}_i$ of instances in one class should be the same Lampert et al. (2014). We simplify $\mathbf{y}_c$ as the semantic prototype for all the instances in class $c$. Given one test instance $\mathbf{x}_i$, our goal is to predict its class label $c_i$. We discuss two tasks: (1) Open set recognition: $c_i \in \{\mathcal{C}_s, 'novel\ class'\}$; (2) Generalized zero-shot learning: $c_i \in \{\mathcal{C}_s, \mathcal{C}_t\}$. The semantic prototype is predefined for each class in $\mathcal{C}_t$. The '*novel class*' is an umbrella term referring to any class not in $\mathcal{C}_s$.

### 3.2 EXTREME VALUE DISTRIBUTIONS

We firstly introduce the background of modeling the extreme values (*i.e.*, minimum/maximum scores) computed from one supervised classifier as the extreme value distributions. In particular, by using each source class $c$, we can train a binary predictor function, *e.g.* SVM, $z^c = f_c(\mathbf{x}) : \mathbb{R}^n \to \mathbb{R}$ where $z$ is the confidence score of instance $\mathbf{x}$ belonging to the class $c$. In Extreme Value Theory (EVT) , the extreme values (i.e., maximum / minimum confidence scores) of the score distribution computed by the predictor function $f_c(\cdot)$ can be modeled by an EVT distribution. Specifically, for instance set $\{\mathbf{x}\}$ that belong to class $c$; the minimum score $z_{\min}^c = \min f_c(\{\mathbf{x}\})$ follows the Weibull distribution,

$$z_{\min}^c \xrightarrow{\mathcal{D}} G(z; \lambda_c, \nu_c, \kappa_c) \tag{1}$$

where $G(\cdot)$ is the Cumulative Distribution Function (CDF) of Weibull distribution ($z > \nu$): $G(z^c; \lambda_c, \nu_c, \kappa_c) = 1 - \exp\left(-\left(\frac{z^c - \nu_c}{\lambda_c}\right)^{\kappa_c}\right)$. Critically, Eq (1) models the lower boundary distribution of instance confidence scores belonging to class $c$. On the other hand, for instance set $\{\mathbf{x}\}$ NOT belonging to class $c$, the maximum score $z_{\max}^c = \max f_c(\{\mathbf{x}\})$ should follow the reverse Weibull distribution Scheirer et al. (2011; 2012; 2014),

$$z_{\max}^c \xrightarrow{\mathcal{D}} rG\left(z; \lambda_c', \nu_c', \kappa_c'\right) \tag{2}$$

where $rG(\cdot)$ is the CDF of reverse Weibull distribution:

$$rG\left(z; \lambda_c', \nu_c', \kappa_c'\right) = 1 - G\left(z; \lambda_c', \nu_c', \kappa_c'\right). \tag{3}$$

Eq (2) models the upper boundary distribution of confidence scores NOT belonging to class $c$. The scale parameters $\lambda_c, \lambda_c'$, shape parameters $\kappa_c, \kappa_c'$, and location parameters $\nu_c, \nu_c'$ are estimated by Maximum Likelihood Estimator fitted from the training data. Critically, Eq. (2) models the upper boundary distribution of instance confidence scores NOT belonging to class $c$.

### 3.3 WEAKNESS OF CATEGORIZING INSTANCES INTO KNOWN/UNKNOWN DOMAINS

One can conduct the OSL and G-ZSL by directly learning to separate the domains of seen and novel/unseen classes. The idea of learning to divide testing instances to known or unknown domains

has been studied in W-SVM Scheirer et al. (2014) (OSL), and CMT Socher et al. (2013) (G-ZSL). Particularly, given a test instance $\mathbf{x}_i$, the supervised binary algorithm can compute the confidence score $z_i^c = f_c(\mathbf{x}_i)$ belonging to the class $c$. Here we introduce two events:

$$E_1 : \mathbf{x}_i \text{ belonging to class c};$$

$$E_2 : \mathbf{x}_i \text{ belonging to the other seen classes.}$$

The distributions of extreme values defined in Eq (1) and Eq (2) can actually give us the boundary of each event above happened in a probabilistic perspective. Thus we can have the probability $P_G(E_1) = 1 - \exp\left(-\left(\frac{z_i^c - \nu_c}{\lambda_c}\right)^{\kappa_c}\right)$; and $P_{rG}(\neg E_2) = \exp\left(-\left(\frac{z_i^c - \nu_c'}{\lambda_c'}\right)^{\kappa_c'}\right)$. Thus, to determine if one testing instance $\mathbf{x}_i$ belong to class $c$, W-SVM Scheirer et al. (2014) computes the statistic $m_c(\mathbf{x}_i)$ by,

$$m_c(\mathbf{x}_i) = P_{rG}(\neg E_2 \mid f_c(\mathbf{x}_i)) \cdot P_G(E_1 \mid f_c(\mathbf{x}_i)). \tag{4}$$

W-SVM introduces a threshold $\delta_c$ to determine whether the instance $i$ belongs to the class $c$ as,

$$c_i = \begin{cases} c & m_c(\mathbf{x}_i) > \delta_c \\ \neg c & \text{otherwise} \end{cases} \tag{5}$$

where $\delta_c$ is a fixed value Scheirer et al. (2014). The instance $\mathbf{x}_i$ rejected by all the seen classes by Eq (5) is labeled as the unknown domain. Generalizing to $\mathcal{C}_s$ class is straightforward by training multiple prediction functions $\{f_c(\mathbf{x})\}, c = 1, \cdots, |\mathcal{C}_s|$.

However, there are several key limitations in directly utilizing Eq (4) and Eq (5) of learning the division of domains: (1) Eq (4) directly multiplies two terms and this indicates that there is a potential hypothesis that no correlation exists between $E_1$ and $\neg E_2$, which is generally not the case. (2) In multiple seen classes, the instances may derive from many different classes. It is hard to determine a single fixed $\delta$ in Eq (5) for each class. (3) Furthermore, we give an illustration of the non-negligible overlapping between the intra-class and inter-class distances of each pair of instances on the aPY dataset (described in Sec. 6.1,Farhadi et al. (2009)). As shown in the feature space of Fig. 2 (b), we compute the pairwise $L_2$ distances over (1) the instances within the same classes (intra-class), and (2) the instance from different classes (inter-class) on aPY dataset Farhadi et al. (2009). We use the empirical density to show the results in Fig. 2 (c). Practically, it is hard to predict the class labels of instances of the overlapped region in the known/unknown domain. Due to the large overlapped region, the instances (*e.g.*, CMT in Socher et al. (2013)) whose domains are wrongly labeled will never be correctly categorized.

## 4  PROPOSED DOMAIN DIVISION ALGORITHM

### 4.1  DETERMINING THE INITIAL BOUNDARY BY BOOTSTRAPPING

The W-SVM in Eq (5) and Eq (4) estimates the confidence scores by a fixed threshold empirically per-defined for any data distributions in the known domain. However, intrinsically, it can be taken as a model selection task in estimating the boundary by Eq (4). In this paper, we tackle the question of constructing the boundary of the known domain via the bootstrap approach Efron (1979). The bootstrapping[2] is a strategy of estimating the standard errors and the confidence intervals of parameters when the underlying distributions are unknown. Its procedures are closely related to the other methods such as cross-validation, and jackknife.

Bootstrapping is the most widely used tool in approximating the sampling distributions of test statistics and estimators. To facilitate the discussion, we denote the training set of class $c$ as $\{\mathbf{x}_{tr}^c\}$; the testing set whose instances are mostly confidently predicted as class $c$, as $\{\mathbf{x}_{te}^c\}$. Thus the corresponding confidence score set on training and testing data are $\{z_{tr}^c\} = f_c(\{\mathbf{x}_{tr}^c\})$ and $\{z_{te}^c\} = f_c(\{\mathbf{x}_{te}^c\})$ respectively. The whole algorithm is shown in Alg. 1.

Till now, we had a sketch of our domain division algorithm. Specifically, the training instances of seen classes are utilized to learn the $f_c(\cdot)$, $c \in \mathcal{C}_s$; For any given testing instance $\mathbf{x}_i$, we compute

---

[2]It is different from "bootstrapping" (i.e., self-training) in computer vision Shrivastava et al. (2012).

---

**Algorithm 1** Determining the initial threshold by bootstrapping

---

**Input:** Confidence score set on training data $\{z_{tr}^c\}$

**Output:** Threshold $\delta_c$:

1. We sample from $\{z_{tr}^c\}$ for $n$ times (with replacement), producing a sampling set $\left\{\widetilde{z}_{tr(k)}^c\right\}_{k=1}^n$, where $\widetilde{z}_{tr(k)}^c$ indicates the $k_{th}$ sampled instance;

2. We also choose the significance level $\alpha$, and generate the $\alpha$ quantile $\widetilde{z}_{tr}^{c\star}$ from $\left\{\widetilde{z}_{tr(k)}^c\right\}_{k=1}^n$. Particularly, we sort $\widetilde{z}_{tr(k)}^c$ with an ascending order and extract $(\max{(\text{Round}\,[\alpha n]\,,1)})_{th}$ value as $\widetilde{z}_{tr}^{c\star}$. We repeat it for $n$ times over $\{z_{tr}^c\}$ to get $\left\{\widetilde{z}_{tr(k)}^{c\star}\right\}_{k=1}^n$.

3. The threshold of Eq (5) can thus be computed as the mean of these values, *i.e.*, $\delta_c = \frac{1}{n}\Sigma_{k=1}^n \widetilde{z}_{tr(k)}^{c\star}$.

---

its confidence score $f_c\left(\mathbf{x}_i\right)$, $c \in \mathcal{C}_s$. To determine whether the class of a instance $\mathbf{x}_i$ is seen or unseen, we calculate the statistic $m_c\left(\mathbf{x}_i\right)$ in Eq (4) and Eq (5) with the threshold $\delta_c$ estimated by the bootstrapping algorithm in Alg. 1. The instances computed in the known and unknown domain will be categorized by supervised, or zero-shot classifiers respectively.

There are still two difficulties in the above framework. (1) The whole framework relies on the classifier $f_c\left(\cdot\right)$, $c \in \mathcal{C}_s$ which is supposed to be robust and well-trained. However, empirically, we can not always train good classifiers for all classes. For example, some class has small number of labeled training instances which are insufficient in training the classifier; some outliers may affect the predictor; the hyper-parameters of the classifiers are wrongly tuned. (2) The naive bootstrapping in Alg. 1 generally provides the bad approximations of the distributions of empirical quantiles in practice Falk & Reiss (1989). Practically, in our tasks, we observe that the estimated $\delta_c$ may be consistently too relaxed to determine the boundary of the known domain. We illustrate such a phenomenon in Figure 2(a): the low-density of seen class bicycle instances (blue points) in the northwest part extends the decision boundary. The relaxed boundary could inadvertently classify unseen instances (red points) as the false positives. Unfortunately, in the framework above, once one testing instance in unseen class is wrongly labeled as the known domain, this instance will never be correctly classified. To address these two problems, we suggest a shrinking step in updating the initial boundary of bootstrapping in the next subsection.

## 4.2 Shrinking the boundary of the known domain by Kolmogorov-Smirnov Test

The key idea of updating initial boundary of bootstrapping is to validate whether the learned classifier $f_c\left(\cdot\right)$, $c \in \mathcal{C}_s$ is trustworthy. Generally, assume the instances of class $c$ independent and identically distributed and provided training samples sufficient, a ideal classifier $f_c\left(\cdot\right)$ should produce the similar confidence score distributions of training and testing instances of class $c$.

The Kolmogorov-Smirnov (K-S) test is an efficient, straightforward, and qualified choice method for comparing distributions Massey Jr (1951); Miller (1956); Wang et al. (2003). Remarkably, K-S test is a distribution free test, and the statistics of K-S test is effortless to compute. We define the null and alternative hypothesis as

$$\begin{aligned} \text{H}_0 &: \{\text{z}_{\text{tr}}^{\text{c}}\} \text{ and } \{\text{z}_{\text{te}}^{\text{c}}\} \text{ are from the same distribution.} \\ \text{H}_1 &: \{\text{z}_{\text{tr}}^{\text{c}}\} \text{ and } \{\text{z}_{\text{te}}^{\text{c}}\} \text{ are from different distributions.} \end{aligned} \tag{6}$$

We introduce a distance measure $K^c = \sup_z \|F_{tr}^c\left(z\right) - F_{te}^c\left(z\right)\|$ where $F_{tr}^c = \text{ecdf}(\{z_{tr}^c\})$ and $F_{te}^c = \text{ecdf}(\{z_{te}^c\})$; and the $\text{ecdf}\left(\cdot\right)$ is the empirical distribution function. The null hypothesis would be rejected at the significant level $\alpha$ when,

$$K^c(\alpha) > \sqrt{-\frac{|\{z_{tr}^c\}| + |\{z_{te}^c\}|}{2\,|\{z_{tr}^c\}| \cdot |\{z_{te}^c\}|}\log\left(\frac{\alpha}{2}\right)}. \tag{7}$$

When $H_0$ is accepted, it indicates that the $f_c\left(\cdot\right)$ is trustworthy, and the confidence scores of training and testing instances in class $c$ come from the same distribution. We are certain that a large portion of testing instances $\{z_{te}^c\} = f_c\left(\{\mathbf{x}_{te}^c\}\right)$ should be indeed in the class $c$. On the other hand, when

$H_0$ is rejected, we are not sure whether $f_c(\cdot)$ is well learned; and the class labels of these testing instances are uncertain. To this end, we introduce a new domain – uncertain domain to include these instances.

**Uncertain Domain.** The labels of instances in the uncertain domain should be labeled as the most likely seen class, or one of unseen classes. Specifically, we can compute the $\{z^c = f_c(\mathbf{x})\}_{c=1}^{|\mathcal{C}_s|}$ over all $\mathcal{C}_s$ classes; and we can obtain,

$$\{c^\star, z^\star\} = \mathrm{argmax}_{c \in \mathcal{C}_s} \{z^c\}. \tag{8}$$

The mapping function $g(\cdot)$ is learned on the known domain from features $\mathbf{x}_i$ to its corresponding semantic attributes $\mathbf{y}_i$. Given one testing instance $\mathbf{x}_i$: if $z_i^\star$ is very high, we can confidently predict $\mathbf{x}_i$ belonging to one of seen classes; otherwise, the label of $\mathbf{x}_i$ is either in the uncertain or unknown domain. We thus have,

$$c_i^\star = \begin{cases} \mathrm{argmax}_{c \in \mathcal{C}_s} f_c(\mathbf{x}_i) & \textit{seen domain} \\ \mathrm{argmin}_{c \in \mathcal{C}_t} \|g(\mathbf{x}_i) - \mathbf{y}_c\| & \textit{unseen domain} \\ \mathrm{argmin}_{c \in \mathcal{C}_t \cup \{c^\star\}} \|g(\mathbf{x}_i) - \mathbf{y}_c\| & \textit{uncertain domain} \end{cases} \tag{9}$$

where $\mathbf{y}_c$ is semantic prototype of class $c$; $c^\star$ is the most likely known class to which $\mathbf{x}_i$ belongs to. Note that in OSL, we only know the $\mathbf{y}_c$ ($c \in \mathcal{C}_s$) of seen classes; We can dynamically construct a $\mathcal{C}_t$ set by randomly generating $\mathbf{y}_i$ by making sure $\|\mathbf{y}_i - \mathbf{y}_j\| > \epsilon$ ($\forall \mathbf{y}_j \in \mathcal{C}_s$), and $\epsilon = \min_{\mathbf{y}_i, \mathbf{y}_j \in \mathcal{C}_s; i \neq j} \|\mathbf{y}_i - \mathbf{y}_j\|$. The sample size is usually the same with the number of target classes.

## 5 Recognition models in known/unknown/uncertain domains

We can apply different recognition algorithms in each domain. In known domain, the standard supervised classifiers can be learned and applied. In unknown and uncertain domains, we propose a simple yet effective feature prototype embedding recognition algorithm as our plain implement.

**Feature prototype embedding.** Once the domain is well separated, we can use the ZSL algorithms to set up the mapping from feature space to semantic/attribute space. In order to confirm that our main contribution is the domain division part, we do not use very complicated ZSL algorithms. Only the simplest linear predictor is utilized here to recognize the unseen classes. Particularly, we use feature prototypes to replace all the instances of each class to avoid the unbalance sample size among classes. We learn a linear predictor to predict the attribute/word vector $g(\mathbf{x}) = \mathbf{w}^T \cdot \mathbf{x}$. The feature prototype embedding is computed as,

$$\mathbf{w} = \mathrm{argmin}_{\mathbf{w}} \sum_{c \in \mathcal{C}_s} \|g(\widetilde{\mathbf{x}}_c) - \mathbf{y}_c\| + \lambda \|\mathbf{w}\|_2 \tag{10}$$

where $\widetilde{\mathbf{x}}_c = \frac{1}{n_s^c} \Sigma_{l_i=c} \mathbf{x}_i$ is the feature prototype of class $c$; $\mathbf{y}_c$ is the semantic prototype of class $c$.

When we tackle an instance in the unknown or uncertain domain, we need to embed features into semantic space with $g$, which can infer the class labels of instances:

$$c_i = \begin{cases} \mathrm{argmin}_{c \in \mathcal{C}_t} \|g(\mathbf{x}_i) - \mathbf{y}_c\| & \textit{unknown domain} \\ \mathrm{argmin}_{c \in \mathcal{C}_t \cup \{c^\star\}} \|g(\mathbf{x}_i) - \mathbf{y}_c\| & \textit{uncertain domain} \end{cases} \tag{11}$$

where $c^\star$ is the most likely seen class for $\mathbf{x}_i$ which is computed by the supervised classifier and $\mathbf{y}_c$ is the semantic prototype.

Also, the experiment in each domain can be done with ANY other ZSL. For instance, we report the implement with *f-CLSWGAN (f-C)* Xian et al. (2018) so that G-ZSL can be done with *f-CLSWGAN* within a single domain.

# 6 EXPERIMENTS

## 6.1 DATASETS AND SETTINGS

**Datasets.** *Animal with Attribute (AwA)* Dataset Lampert et al. (2014) has 50 classes and 30,475 images in total, with 85 class-level attributes annotated. We use 40 source training classes (including 13 classes as validation); the rest as testing. *CUB* Dataset Wah et al. (2011) includes 200 classes and 11,788 fine-grain images with 312 class-level attributes annotated. The training set has 150 classes (including 50 classes as validation). (3) *aPY* Dataset Farhadi et al. (2009) has 15,339 images in 32 classes with 64 class-level annotated attributes. We use 20 classes for training (including 5 validation classes). For the AwA, CUB and aPY, we use ResNet-101 features and the class split contributed by Xian et al. (2017). (4) ImageNet 2012/2010 dataset is proposed in Fu & Sigal (2016). As the large-scale dataset, we use the split as Fu & Sigal (2016): 1000 training classes with full training instances in ILSVRC 2012; and 360 testing classes in ILSVRC 2010, non-overlapped with ILSVRC 2012 classes. We address two recognition tasks: OSL and G-ZSL.

**Experimental settings.** Our model is validated in OSL and G-ZSL settings. OSL identifies whether an image belongs to the one of seen classes or the novel class. G-ZSL gives the class label of testing instances either from seen or unseen classes. We set the significance level $\alpha = 0.05$ to tolerate 5% Type-I error. By default, we use SVM with RBF kernel with parameter cross-validated, unless otherwise specified.

## 6.2 RESULTS OF OPEN SET LEARNING

We compare against the competitors, including Attribute Baseline (Attr-B), W-SVM Scheirer et al. (2014), One-class SVM SchÄlkopf et al. (2001), Binary SVM, OSDN Bendale & Boult (2016) and LOF Breunig et al. (2000). The attribute baseline is the variant of our task without using domain division algorithm. Particularly, the Attr-B uses the same semantic space and embedding as our model, but does not leverage domain division step, *i.e.*, use negative samples and prototypes to identify projected instances directly (Fig. 1 (c)).

We use the metric – F1-measure, which is defined as the harmonic mean of seen class accuracy (specific class) and unseen prediction accuracy (unnecessary to predict the specific class). The results are compared in Tab. 1. Significant performance gain over existing approaches has been observed, in particular for AwA, aPY and ImageNet. This validates the effectiveness of our framework. We attribute the improvement to the newly introduced uncertain domain which help better differentiate whether testing instances derive from known or unknown domain.

Table 1: Comparison of open set recognition algorithms

| Method / Accuracy | AwA | CUB | aPY | ImageNet |
|---|---|---|---|---|
| Attr-B | 33.8 | 18.7 | 5.1 | 3.7 |
| Binary SVM | 57.7 | 29.8 | 66.6 | 24.6 |
| W-SVM | 80.2 | 58.6 | 78.6 | 50.1 |
| One-Class SVM | 58.9 | 27.6 | 57.1 | 23.4 |
| OSDN | 49.9 | 36.7 | 41.5 | – |
| LOF | 60.0 | 54.5 | 49.1 | 38.0 |
| Ours | **93.7** | **59.5** | **94.3** | **67.6** |

## 6.3 RESULTS OF GENERALIZED ZERO-SHOT LEARNING

**Settings**: We first compare the experiments on G-ZSL by using the settings in Xian et al. (2017). The results are summarized in Tab. 2. In particular, we further compare the separate settings; and top-1 accuracy in (%) is reported here: (1) $\mathbb{S} \to \mathbb{T}$: Test instances from seen classes, the prediction candidates include both seen and unseen classes; (2) $\mathbb{U} \to \mathbb{T}$: Test instances from unseen classes, the prediction candidates include both seen and unseen classes. (3) We employ the harmonic mean as the main evaluation metric to further combine the results of both $\mathbb{S} \to \mathbb{T}$ and $\mathbb{U} \to \mathbb{T}$, as $H = 2 \cdot (Acc(\mathbb{U} \to \mathbb{T}) \times Acc(\mathbb{S} \to \mathbb{T})) / (Acc(\mathbb{U} \to \mathbb{T}) + Acc(\mathbb{S} \to \mathbb{T}))$.

**Competitors**. We compare several competitors. (1) *DAP* Lampert et al. (2014), trains a probabilistic attribute classifier and utilizes the joint probability to predict labels; (2) *ConSE* Norouzi et al. (2013), maps features into the semantic space by convex combination of attributes; (3) *CMT* Socher et al.

Table 2: G-ZSL Results on AwA, CUB and aPY. (* Our implement; [1] Plain implement: prototype linear mapping; [2] Implement with f-C)

| | AwA | | | CUB | | | aPY | | |
|---|---|---|---|---|---|---|---|---|---|
| | $\mathbb{U} \to \mathbb{T}$ | $\mathbb{S} \to \mathbb{T}$ | $H$ | $\mathbb{U} \to \mathbb{T}$ | $\mathbb{S} \to \mathbb{T}$ | $H$ | $\mathbb{U} \to \mathbb{T}$ | $\mathbb{S} \to \mathbb{T}$ | $H$ |
| Chance | 2.0 | 2.0 | - | 0.5 | 0.5 | - | 3.1 | 3.1 | - |
| DAP | 0.0 | 88.7 | 0.0 | 1.7 | 67.9 | 3.3 | 4.8 | 78.3 | 9.0 |
| ConSE | 0.4 | 88.6 | 0.8 | 1.6 | 72.2 | 3.1 | 0.0 | **91.2** | 0.0 |
| CMT | 8.4 | 86.9 | 15.3 | 4.7 | 60.1 | 8.7 | 10.9 | 74.2 | 19.0 |
| SSE | 7.0 | 80.6 | 12.9 | 8.5 | 46.9 | 14.4 | 0.2 | 78.9 | 0.4 |
| Latem | 7.3 | 71.7 | 13.3 | 15.2 | 57.3 | 24.0 | 0.1 | 73.0 | 0.2 |
| ALE | 16.8 | 76.1 | 27.5 | 23.7 | 62.8 | 34.4 | 4.6 | 73.7 | 8.7 |
| DeViSE | 13.4 | 68.7 | 22.4 | 23.8 | 53.0 | 32.8 | 4.9 | 76.9 | 9.2 |
| SJE | 11.3 | 74.6 | 19.6 | 23.5 | 59.2 | 33.6 | 3.7 | 55.7 | 6.9 |
| ESZSL | 6.6 | 75.6 | 12.1 | 12.6 | 63.8 | 21.0 | 2.4 | 70.1 | 4.6 |
| SYNC | 8.9 | 87.3 | 16.2 | 11.5 | **70.9** | 19.8 | 7.4 | 66.3 | 13.3 |
| SAE | 1.1 | 82.2 | 2.2 | 7.8 | 54.0 | 13.6 | 0.4 | 80.9 | 0.9 |
| SE-G | 56.3 | 67.8 | 61.5 | 41.5 | 53.3 | 46.7 | - | - | - |
| cycle-C | 56.9 | 64.0 | 60.2 | 45.7 | 61.0 | 52.3 | - | - | - |
| f-C | 57.9 | 61.4 | 59.6 | 43.7 | 57.7 | 49.7 | - | - | - |
| PTMCA | 22.4 | 80.6 | 35.1 | 23.0 | 51.6 | 31.8 | 15.4 | 71.3 | 25.4 |
| f-C* | 57.8 | 72.4 | 64.2 | 43.4 | 58.3 | 49.8 | 16.8 | 45.7 | 24.6 |
| Ours[1] | 53.6 | 90.4 | 67.3 | 37.2 | 45.2 | 40.8 | **44.0** | 89.2 | **58.9** |
| Ours[2] | **66.0** | **91.2** | **76.6** | **53.1** | 59.4 | **56.1** | 22.4 | 81.3 | 35.1 |

(2013), projects features into unsupervised semantic space and uses LOF to detect novel classes; (4) *SSE* Zhang & Saligrama (2015), regards novel classes as mixtures of seen proportions to measure the instance similarity. (5) *Latem* Xian et al. (2016), is a novel latent embedding for ZSL and G-ZSL. (6) *ALE* Akata et al. (2016), embeds labels into the attribute space by learning a function to rank the likelihood of each class. (7) *DeViSE* Frome et al. (2013), uses both unsupervised information and annotated attributes to classify classes in an embedding model; (8) *SJE* Akata et al. (2015) is a hierarchical embedding to learn an inner product gram matrix between features and attributes. (9) *ESZSL* Romera-Paredes & Torr (2015), focuses on the regularization term in the projection from features to semantic space. (10) *SYNC* Changpinyo et al. (2016), aligns the semantic space to feature space by manifold learning. (11) *SS-VOC* Fu & Sigal (2016), optimizes the triplet loss to learn the projection from features to semantic space. (12) *SAE* Kodirov et al. (2017) is an auto-encoder to combine feature and semantic space. (13) *SE-GZSL* Verma et al. (2018) leverages VAE Kingma & MaxWelling (2014) as the generator of pseudo instances to train the mapping. (14-15) *cycle-CLSWGAN (cycle-C) & f-CLSWGAN (f-C)* Felix et al. (2018); Xian et al. (2018): both of them use W-GAN Arjovsky et al. (2017) to reconstruct features and cycle-C adds another regularizer. (16) *PTMCA* Long et al. (2018) uses transfer approach and embedding way to reduce bias and variance.

Table 3: G-ZSL on the large-scale dataset – ImageNet 2012/2010.

| | SS-Voc | SAE | ESZSL | DeViSE | ConSE | Chance | Ours |
|---|---|---|---|---|---|---|---|
| $\mathbb{U} \to \mathbb{T}$ | 2.3 | 0.2 | 0.5 | 0.4 | 0.0 | <0.1 | **5.7** |
| $\mathbb{S} \to \mathbb{T}$ | 33.5 | 32.8 | 38.1 | 24.7 | **56.2** | <0.1 | 54.1 |
| $H$ | 4.3 | 0.5 | 0.9 | 0.8 | 0.0 | - | **10.3** |

**Results**. As seen in Tab. 2, our harmonic mean results are significantly better than all the competitors on almost all the datasets. This shows that ours can effectively address the G-ZSL tasks. Particularly,

(1) Our plain results can beat other competitors by a large margin on AwA and aPY dataset, due to the efficacy of our domain division algorithm. Also, thanks to the power of *f-CLSWGAN (f-C)*, our implement with it on both CUB and AwA dataset are impressive. (2) The key advantage of the proposed framework is learning to better divide the testing instances into *known*, *uncertain* and *unknown* domains. In the known domain, we use the standard SVM classifier. In unknown/uncertain

Table 4: Ablation Study. $\sqrt{}$/$\times$ indicate *using*/*not using* the corresponding step respectively.

| Dataset | AwA | | | | aPY | | | | CUB | | | |
|---|---|---|---|---|---|---|---|---|---|---|---|---|
| K-S test | $\sqrt{}$ | $\sqrt{}$ | $\times$ | $\times$ | $\sqrt{}$ | $\sqrt{}$ | $\times$ | $\times$ | $\sqrt{}$ | $\sqrt{}$ | $\times$ | $\times$ |
| Bootstrap | $\sqrt{}$ | $\times$ | $\sqrt{}$ | $\times$ | $\sqrt{}$ | $\times$ | $\sqrt{}$ | $\times$ | $\sqrt{}$ | $\times$ | $\sqrt{}$ | $\times$ |
| OSL | **93.7** | 85.6 | 37.1 | 80.2 | **94.3** | 85.1 | 36.8 | 78.6 | **59.5** | 59.3 | 32.9 | 58.6 |
| G-ZSL | **67.3** | 63.5 | 11.4 | 61.7 | **58.9** | 40.5 | 6.9 | 19.5 | **40.8** | 38.1 | 12.1 | 31.0 |

domains, we directly embed feature prototypes into semantic space and match the most likely class in the candidate pool. This is the most simple and straightforward recognition method. Thus our good harmonic mean performance on G-ZSL largely comes from the good domain division algorithm. Additionally, we also highlight that the other advanced supervised or zero-shot algorithms are orthogonal and potentially be useful in each domain if we want to further improve the performance of G-ZSL. (3) Our framework is also applied to large-scale datasets in Tab. 3. We compare several state-of-the-art methods that address G-ZSL on the large-scale dataset. We use the SVM with the linear kernel on this dataset, due to the large data scale. Our harmonic mean results surpass the other competitors with a very significant margin. We notice that other algorithms have very poor performance on $\mathbb{U} \to \mathbb{T}$. This indicates the intrinsic difficulty of G-ZSL on large-scale dataset. In contrast, our domain division algorithm can better separate the testing instances into different domains; thus achieving better recognition performance. Additionally, we found that the prediction of ConSE Norouzi et al. (2013) is heavily biased towards known classes which is consistent with the results in small datasets. This is due to the probability of unseen classes are expressed as the convex combination of seen classes. Usually, there is no higher probability would be assigned to unseen classes than the most probable seen class, especially for large datasets.

## 6.4 ABLATION STUDY

In the ablation study, we report the F1-measure and Harmonic mean for OSL and G-ZSL respectively with our plain implement. As is illustrated in Fig. 2, we notice that although the distance statistic shows the different histogram patterns in feature space, the overlapping part is not negligible.

**Importance of bootstrapping the initial threshold**. We introduce a *variant A* of our framework by replacing bootstrapping step (Sec. 4.1) by using Eq (4) and Eq (5) to fix the threshold (*i.e.*, W-SVM Scheirer et al. (2014)), *i.e.*, K-S test ($\sqrt{}$), and Bootstrap ($\times$). As in Tab. 4, the results of *variant A* are significantly lower than our framework on all three datasets. This actually directly validates the importance of determining the initial threshold by bootstrapping.

**Improvements of fine-tuning the threshold by K-S test.** We define the *variant B* is to only use step without fine-tuning the boundary by K-S Test (in Sec. 4.2). Table 4 directly shows the improvement with/without fine-tuning the threshold, *i.e.*, K-S test ($\times$), and Bootstrap ($\sqrt{}$). In particular, we note that *variant B* has significant lower results on OSL and G-ZSL than *variant A* and our framework. One reason is that our bootstrapping step actually learns to determine a very wide boundary of the known domain, to make sure the good results in labeling testing instances as unknown domain samples. The fine-tuning threshold step will further split the individual known domain into known/uncertain domain by shrinking the threshold. Without such a fine-tuning step, *variant B* may wrongly categorize many instances from unseen classes as one of the known classes. Thus, we can show that the two steps of our framework are very complementary to each other and they work as a whole to enable the good performance on OSL and G-ZSL. Finally, we introduce the *variant C* in Tab. 4, by using W-SVM to do OSL, and then use our ZSL model for G-ZSL, *i.e.*, K-S test ($\times$), and Bootstrap ($\times$). The performance of *variant C* is again significantly lower than that of ours, and this demonstrates the efficacy of our model.

## 7 CONCLUSION

This paper learns to divide the instances into *known, unknown and uncertain* domains for the recognition tasks from a probabilistic perspective. The domain division procedure consists of bootstrapping and K-S Test steps. The bootstrapping is used to set an initial threshold for each class; we further employ the K-S test to fine-tune the boundary. Such a domain division algorithm can be used for OSL and G-ZSL tasks, and achieves remarkable results.

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
