# OpenReview forum: "Learning to Separate Domains in Generalized Zero-Shot and Open Set Learning: a probabilistic perspective"
_ICLR.cc/2019/Conference_

### Official Review · AnonReviewer1 · 2018-10-19
**Good but lengthty paper; approach seems solid and results are generally convincing but could show more comparisons and examples**

**Rating:** 6
**Confidence:** 3

**Review:**

This paper proposes to introduce a new domain, the uncertain domain, to better handle the division between seen/unseen domains in open-set and generalized zero-shot learning. The approach handles test samples estimated to be from the seen class in one way, and ones that belong to either the unseen or uncertain domain in another way. This idea handles the problem that test instances may incorrectly be attributed to one of the seen classes. The authors evaluate their approach on several relevant datasets against a wide variety of methods for OSL and ZSL, and show convincing results.

I have three concerns. One is that the method sections of the paper are fairly lengthy, including an extensive explanation of prior work, e.g. EVT, so time is spent reading before the reader gets to the interesting part of the proposed method, and this time could be better focused around the contributions of *this* work.

For the G-ZSL experiments, most of the methods seem to be older methods tackling ZSL not G-ZSL so perhaps more relevant baselines could be found.

On a related note, it would be good to include some qualitative examples that might reveal some intuitive reasons for the large margin between the performance of the proposed work, and other approaches; in some cases this margin seems rather large, and while the authors attempt to explain it, something beyond a textual explanation might be useful.

---

> ### Author Response · Authors · 2018-11-25
> **RE: Good but lengthty paper; approach seems solid and results are generally convincing but could show more comparisons and examples**
>
> Q1: One is that the method sections of the paper are fairly lengthy, including an extensive explanation of prior work, e.g. EVT, so time is spent reading before the reader gets to the interesting part of the proposed method, and this time could be better focused around the contributions of *this* work.
>
> A1: Thank you for your advice. We have done some minor fix to let it better.
>
> Q2: For the G-ZSL experiments, most of the methods seem to be older methods tackling ZSL not G-ZSL so perhaps more relevant baselines could be found.
>
> A2: We have supplemented the comparison with new G-ZSL algorithms and our predominance is still significant.
>
> Q3: On a related note, it would be good to include some qualitative examples that might reveal some intuitive reasons for the large margin between the performance of the proposed work, and other approaches; in some cases this margin seems rather large, and while the authors attempt to explain it, something beyond a textual explanation might be useful.
>
> A3: Thank you. Actually, since we reduce the candidate set of prediction significantly, the difficulty of G-ZSL reduced a lot. For example, about AwA the size of the full class is 50, but, if we use our algorithm, the size for seen, unseen, uncertain is 40, 10, 11, respectively. Besides, we add another variant to prove the power of such a reduction.

---

> > ### Comment · AnonReviewer1 · 2018-12-03
> > **thanks**
> >
> > Thank you for your response and adjustments. The paper does look better. Not sure which variant in the paper is the variant referenced in the comment above (reduction). I retain my weakly positive rating.

---

> > > ### Author Response · Authors · 2018-12-04
> > > **RE: thanks**
> > >
> > > Thank you for your reply.
> > > The variant we referenced is also the latest version now. I would be very appreciated if you can take a closer look.
> > >
> > > Thanks.

---

### Official Review · AnonReviewer2 · 2018-11-02
**Interesting ideas on domain separation, impressive results on GZSL, but some problems with clarity and comparative evaluation**

**Rating:** 5
**Confidence:** 3

**Review:**

This paper describes an approach to domain separation based on bootstrapping to identify similarity cutoff thresholds for known classes, followed by a Kolmogorov-Smirnoff test to refine the bootstrapped in-distribution zones. The authors apply these techniques to two general recognition problems: open-world recognition, and Generalized Zero-shot Learning (GZSL). Experimental results are given on a variety of datasets, and a thorough comparative evaluation on GZSL is performed.

The paper has the following strong points:

 1. The motivations for each element of the proposed approaches are fairly well presented and are compelling.

 2. The experimental results on GZSL are impressive, especially compared to established approaches.

The paper also has the following weak points:

 1. The writing is a bit rough throughout, though not to extreme distraction. The abstract starts right off with the awkward "This paper studies the problem of domain division problem..." The manuscript needs more careful revision for clarity.

 2. Related to the previous point, there are several elements of the technical exposition that are lacking in clarity. For example, it is unclear what eq. 9 is defining exactly. It seems to be the final decision rule, but it is not clear how seen, unseen, and uncertain samples are determined. Algorithm 1 is clear, but there is never a clear, complete definition of the end-to-end decision pipeline given. I feel like it would be difficult to reproduce the results of this article without significant trial-and-error.

 3. The authors seems to have relied on the Xian et al. paper to extract data for their comparative evaluation on GZSL. There are more recent works from 2018 that should be included, such as:

 Arora, Gundeep, Vinay Kumar Verma, Ashish Mishra and Piyush Rai. “Generalized Zero-Shot Learning via Synthesized Examples." CVPR 2018.

In summary, this paper has many interesting ideas, and the GZSL results are indeed impressive (with respect to the results in the comparison). However, there are many problems with clarity and missing, recent work in the comparative evaluation.

---

> ### Author Response · Authors · 2018-11-25
> **RE: Interesting ideas on domain separation, impressive results on GZSL, but some problems with clarity and comparative evaluation**
>
> Q1: The writing is a bit rough throughout, though not to extreme distraction. The abstract starts right off with the awkward "This paper studies the problem of domain division problem..." The manuscript needs more careful revision for clarity.
>
> A1: Thank you for your advice, we have proofread the paper and fixed such errors.
>
> Q2: Related to the previous point, there are several elements of the technical exposition that are lacking in clarity. For example, it is unclear what eq. 9 is defining exactly. It seems to be the final decision rule, but it is not clear how seen, unseen, and uncertain samples are determined. Algorithm 1 is clear, but there is never a clear, complete definition of the end-to-end decision pipeline given. I feel like it would be difficult to reproduce the results of this article without significant trial-and-error.
>
> A2: Thanks. Based on your comment, we included detailed descriptions for both Alg.1 and Eq (9).  Eq.9 describes how to predict the label of x_i for the seen, unseen, and uncertain domain, which are defined in Section 4.1 and 4.2.
> Regarding the reproduction of the results, we will release the codes if the paper is accepted.
>
> Q3: The authors seem to have relied on the Xian et al. paper to extract data for their comparative evaluation on G-ZSL. There are more recent works from 2018 that should be included, such as Arora, Gundeep, Vinay Kumar Verma, Ashish Mishra, and Piyush Rai. “Generalized Zero-Shot Learning via Synthesized Examples." CVPR 2018.
>
> A3: As suggested, we have compared against several state of the art results, including [Verma et al. 18] you have mentioned. Our method significantly outperforms theirs (See SE-G in Table 2).

---

### Official Review · AnonReviewer3 · 2018-11-04
**a paper with a good potential but difficult to read and missing recent baselines**

**Rating:** 5
**Confidence:** 4

**Review:**

This paper deals with the difficult problem of novelty recognition which is the core issue in open set learning and generalized zero-shot learning. Indeed a method able to separate samples between known and unknown domains in these settings would clearly indicate the direction for their solution. The idea proposed here consists in starting from the extreme value theory and then using bootstrapping to model the confidence score for a sample of belonging or not to a certain class. Through a probabilistic evaluation (based on K-S test) on the trustworthy of each category classifiers, the domain separation is extended to consider also an uncertain domain and the separation threshold is progressively refined. Once the domains are separated, classification can be performed disjointly in each of them.

+ Having a way to define known/unknown and uncertain samples on the basis of which one can
then proceed to solve OSL and GZSL sounds as a very effective strategy. Moreover, all the parts of the method are
based on reliable probabilistic principles.

- Unfortunately the text is not easy to read. There are several repetitions and disordered lists (same numbers used multiple times or mixing names and numbers for the items) which distract the reader. As a side note, it would be better to avoid mentioning dataset names without their description and definition ('aPY' appears out of the blue in the introduction).

- The experiments extends over different datasets and the ablation study is valuable. However to understand how the proposed method advances over the current state of the art it is important to consider and discuss the most recent publications  on OSL and GZSL. See for instance
Open Set Learning with Counterfactual Images, ECCV 2018
Feature Generating Networks for Zero-Shot Learning, CVPR 2018

---

> ### Author Response · Authors · 2018-11-25
> **RE: a paper with a good potential but difficult to read and missing recent baselines**
>
> Q1: Unfortunately the text is not easy to read. There are several repetitions and disordered lists (same numbers used multiple times or mixing names and numbers for the items) which distract the reader. As a side note, it would be better to avoid mentioning dataset names without their description and definition ('aPY' appears out of the blue in the introduction).
>
> A1: Thanks for your advice. We have revised some of these flaws.
>
> Q2: The experiments extend over different datasets and the ablation study is valuable. However to understand how the proposed method advances over the current state of the art it is important to consider and discuss the most recent publications on OSL and G-ZSL. See for instance
> [1] Open Set Learning with Counterfactual Images, ECCV 2018
> [2] Feature Generating Networks for Zero-Shot Learning, CVPR 2018
>
> A2: We have supplemented the comparison with [2] and other new methods. We even add an implement for [2] to improve it. Although the task of [1] seems similar to ours, the purposes are not the same. For our approach and W-SVM, we address the build a boundary in REAL manifolds to detect outliers. However, for OpenMax and [1], they intend to resist attacks from artificial samples. So you can see that the result of OpenMax (OSDN) in our experiment is not very good. Meanwhile, [1] did not release their codes or detailed parameter setting, we are unable to reproduce it in a short period of time.

---

### Public Comment · (anonymous) · 2018-10-19
**Weak comparison**

It is interesting to see the paper is handling more realistic and challenging problem, i.e., G-ZSL. I have two points here:
1- Here it will be more interesting to see the ZSL result also. Can you include the result of ZSL also?
2- The main contribution of the paper are G-ZSL, there is a lot of recent work handling this problem, but the author has not compared with any recent work of G-ZSL. Can you discuss and compare few recent work of G-ZSL, e.g. :
    a- http://openaccess.thecvf.com/content_cvpr_2018/papers/Xian_Feature_Generating_Networks_CVPR_2018_paper.pdf (CVPR-18)
    b- http://openaccess.thecvf.com/content_cvpr_2018/papers/Verma_Generalized_Zero-Shot_Learning_CVPR_2018_paper.pdf (CVPR-18)
    c- http://openaccess.thecvf.com/content_ECCV_2018/papers/RAFAEL_FELIX_Multi-modal_Cycle-consistent_Generalized_ECCV_2018_paper.pdf (ECCV-18)

Thanks

---

> ### Author Response · Authors · 2018-10-20
> **Reply to "Weak comparison"**
>
> Thank you for your comments.
>
> 1. Here it will be more interesting to see the ZSL result also. Can you include the result of ZSL also?
> — As you mentioned, our main contribution is the domain division part. The ZSL result only depends on the mapping we mentioned, which we do not highlight. Thus, we did not report the ZSL results. Actually, the result of this simply mapping is very close to the state-of-the-art (AWA 73.3; aPY 46.8; CUB 39.6) with much fewer computation resources. This kind of simple mapping can prove the efficacy of the domain division algorithm.
>
> 2. The main contribution of the paper are G-ZSL, there is a lot of recent work handling this problem, but the author has not compared with any recent work of G-ZSL. Can you discuss and compare a few recent works of G-ZSL.
> — Admittedly, all of the works you mentioned are very interesting and promising, however, they all synthesize the pseudo-instances in the target domain, which is distinct from previous methods. But these works are complementary and potentially useful to our works. As we said, our mapping based on very simple and classic algorithms, in order to emphasize the meaning of domain division. Moreover, the works you mentioned are also likely to be enhanced by our division algorithm, which reflects the adaptability of our work.
> — Additionally, although our work is not accuracy-oriented, our results are still promising in some datasets, like AWA (Ours: 67.3; Verma et al.: 61.5; Felix et al.: 60.2; Xian et al.: 59.6).

---

> > ### Public Comment · (anonymous) · 2018-10-21
> > **Reply**
> >
> > Thanks for your reply.
> >
> > I found this work is very interesting and appreciate the work, but my point is you have to mention the current state-of-art.
> >
> > 1- Just my point was in the table-2 you have taken all the competitor that results are far below than yours. You have taken max H-mean 27 (ALE) while current state-of-art is ~60.
> > 2- It's true that recent work on the G-ZSL is synthesizing the example, but this is a way of solving the problem, you can't say they are synthesizing the example, and I am not so I will not mention their result.
> > 3- Also in the case of CUB dataset current state-of-art has H-mean ~60, while in the table you have shown the best competitor result ~34.
> >
> > Thanks

---

> > > ### Author Response · Authors · 2018-10-26
> > > **Rely. Thanks! Good points, and some clarification**
> > >
> > > 1， current state-of-art is ~60.
> > > Thanks! We just check several papers you recommended. However, we didnot find the reporte H-mean of 60. It would be very nice if you could point out the paper. We will refere this paper. We also want to emphasize several points,
> > > (1) The whole paper aims at  Learning to Separate Domains in Generalized Zero-Shot and Open Set Learning: a probabilistic perspective. The focus is on domain division algorithm from a probabilistic perspective. So we only use linear model for zero-shot learning parts (rather than design a bit "heavy" zero-shot learning algorithm); and we show that even by using such a simple model, our model still works very well (thanks to well separating the sevearl domains). Of course, designing a good ZSL/G-ZSL is also an extra-ordinary work, which nevertheless would be enough to make an independent paper.
> > > (2) Admittedly, in some datasets, our model has slightly inferior results, and yet our results are generally good on most of dataset. Thus the results show the efficacy of proposed models, since again we actually use very simple forms in the recognition parts.
> > >
> > > 2,Discussion about synthesis instances.
> > > Yes, it's really a brillant point of employing synthesis instances to solve the task, since it seems like a "free lunch". We actually mean that the works of using synthesis instances for Open-Set/ZSL/G-ZSL are orthogonal but potentially useful to our works. Thanks again for mentioning it.

---

> > ### Author Response · Authors · 2018-12-04
> > **updated results**
> >
> > Thanks for the comments. We updated our methods by using non-linear mapping (rather than only using linear mapping) in each divided domain. So our results can get substantially improved. Please kindly check our updated results. Thanks!

---

### Author Response · Authors · 2018-12-09
**Summary of the Revision**

Dear reviewers,

In order to make this work more complete and convincing, we have revised our paper on the following parts:
(1) add the recent state-of-the-art baselines;
(2) implement our algorithm on ``Feature Generating Networks for Zero-Shot Learning’’ to prove the generalization ability of domain division;
(3) fix some typos and errors;
(4) clarify some confusing parts.

Note: The complex part of our paper includes the definitions and derivations based on the extreme value theory. Our goal is to make the paper logical, amenable, and reasonable, so it cannot be very easy to illustrate the core mathematical concept in a short page.

We would be very appreciated if you can take a closer look.

Thank you!

---

### Meta-Review · Area_Chair1 · 2018-12-15
**The idea has some merits.**

**Confidence:** 4
**Recommendation:** Reject

**Metareview:**

AR1 finds the paper overly lengthy and ill-focused on contributions of this work. Moreover, AR1 would like to see more results for G-ZSL. AR2 finds the  paper is lacking in clarity, e.g. Eq. 9, and complete definition of the end-to-end decision pipeline is missing. AR2 points that the manuscript relies on GZSL and comparisons to it but other more recent methods could be also cited:
- Generalized Zero-Shot Learning via Synthesized Examples by Verma et al.
- Zero-Shot Kernel Learning by Zhang et al.
- Model Selection for Generalized Zero-shot Learning by Zhang et al.
- Generalized Zero-Shot Learning with Deep Calibration Network by Liu et al.
- Multi-modal Cycle-consistent Generalized Zero-Shot Learning by Felix et al.
- Open Set Learning with Counterfactual Images
- Feature Generating Networks for Zero-Shot Learning
Though, the authors are welcome to find even more relevant papers in google scholar.

Overall, AC finds the paper interesting and finds the idea has some merits. Nonetheless, two reviewers maintained their scores below borderline due to numerous worries highlighted above. The authors are encouraged to work on presentation of this method and comparisons to more recent papers where possible. AC encourages the authors to re-submit their improved manuscript as, at this time, it feels this paper is not ready and cannot be accepted to ICLR.